# GIS Based Procedural Modeling in 3D Urban Design

**Ming Zhang** [1,2], **Jielin Wu** [1,2], **Yang Liu** [1,2,*], **Ji Zhang** [1,2] **and Guanyao Li** [1,2]

1    Guangzhou Urban Planning & Design Survey Research Institute, Guangzhou 510060, China
2    Guangdong Enterprise Key Laboratory for Urban Sensing, Monitoring and Early Warning,
     Guangzhou 510060, China
*    Correspondence: liuyang@gzpi.com.cn

**Abstract:** Traditional urban design is time-consuming and laborious. We propose a computer-generated architecture (CGA)-based workflow in this work, with the goal of allowing designers to take advantage of a high level of automation. This workflow is based on procedural modeling. A three-step CGA rule was applied to implement 3D urban procedural modeling, (1) parcel subdivision and clustering, (2) building extrusion, and (3) texture mapping. Parcel subdivision and clustering is the key step of layout modeling, giving the modeler flexibility to adjust the placement and size of the inner building lots. Subsequently, a land-use-based combination of eight common building types and layouts was used to generate various urban forms for different urban functional zones. Finally, individual buildings were decorated by creating texture maps of a planar section of the building facade or, alternatively, decomposing facades into sets of repeating elements and texture maps. We employed the proposed workflow in the H-village urban redevelopment program and an air–rail integration zone development program in Guangzhou. Three design proposals were generated for each project. The results demonstrated that this workflow could generate multiple layout proposals and alternative facade textures quickly and, therefore, address most of the collaborative issues with its analysis functions, including a flexible adjustment mechanism and real-time visualization.

**Keywords:** procedural modeling; urban design; layout modeling; building extrusion; texture mapping; Guangzhou

## 1. Introduction

The main incentives for adopting generative design are to use computational capabilities to support human designers and automate parts of the design process [1]. However, it is less applied in urban design than in urban planning and architecture. Urban design "includes technical questions of urban functioning, economic issues of cost and benefit, aesthetic issues of appearance, as well as social issues involving allocation and provision" [2], which differentiates urban design from architecture design and urban planning. Urban design generally involves multiple stakeholders, often representing conflicting requirements and interests, thus intensifying the complexity of the design [3]. Hence, it is necessary for designers to come up with alternative design solutions, present in different formats, and quickly respond to the pop-up requirements. Compared with urban planning, urban design focuses more on design and user experience, and it operates at the feature and system level. Bringing the designer's concept to the non-design expert's communicative level requires significant application of communication media [4]. Thus, collaboration, communication, and visualization are at the heart of urban design.

With advances in computer science and visualization, computer-assisted design has been gradually replacing routine design with fully or semi-automated design procedures. However, studies on creative design issues remain elusive [5], as with collaborative issues. In urban design, the routine design issues are mainly related to the zoning (or detailed plan in China) regulation. That is to say, the design can only be created with certain functions and forms in the way that a detailed plan allows. The functions are defined by land use,

while the forms are limited by parameters such as floor area ratio, density, and height in a detailed plan. Since an individual function can be translated into several pattern languages, which are common to all urban designers, it can theoretically be learned well by a computer, making it possible to generate alternative patterns that follow the parameters. Even though it is probably unrealistic to solve all the design issues with design automation, a collective automation tool still offers extra support and advantages for human designers. The ideal tool should consist of optimized, automated layout design methods, real-time visualization, a mechanism to adapt the design to programmatic changes, and so on.

There have been some enlightening research works on constructing computer-assisted frameworks and models to aid designers in urban design in recent years. These works can be broadly divided into: (1) geometrical modeling, which focuses on algorithms that produce intricate geometry quickly from a compact set of specifications (i.e., procedural modeling) [6,7], and (2) behavioral modeling, the direction of which is to understand the underlying socioeconomic, meteorological, and resource consumption/waste production processes occurring within an urban space [8]. To achieve an optimized urban design, one of the prerequisites is a thorough understanding of the underlying logic of the design which relies on both geometrical modeling and behavioral modeling. It is especially true that geometrical modeling plays a dominant role during the design stage [9–11]. The present study puts forward a collective solution regarding procedural modeling and outlines its application in projects to explore its future potential.

The main contribution of this study is the development of a complete interactive workflow for semi-automated 3D urban design from the 2D representation of the urban area. This workflow links the main urban planning parameters with the 3D urban design model so that the urban design scheme can be changed on the fly according to the requirements of urban planning and the ideas of urban designers. As a result, this workflow greatly enhances collaboration amongst urban designers and significantly boosts their creativity.

## 2. Related Work

Our method builds upon previous work in procedural modeling [12–14]. Procedural modeling is often used to create objects with a high degree of redundancy. It is developed upon some production systems such as L-systems, shape grammars, and split grammars which allow the creation of complex structures from small sets of inputs [15–17]. The L-system was proposed by Lindermayer as a basis for geometric plant modeling [18]. Parish and Müller [7] introduced L-systems to resemble the growth of streets. Shape grammars, which define rules for the specification and transformation of 2D and 3D shapes, were initially used for describing geometric shapes in artworks [19,20]. Wonka et al. [17] extended the concept of shape grammars with split grammars by adding attributes as parameters to the geometric shape itself. Split grammars provide an automated size-independent approach to derive building models from a dataset of rules and attributes. Following these novel approaches, Müller et al. [21] proposed a computer-generated architecture (CGA) method for the generation of detailed 3D objects, particularly urban objects such as buildings and roads. CGA has become one of the most powerful methodologies for procedural modeling and urban design with a set of shape grammars, such as extrusion, translation, scaling, and splitting.

A series of new methods have been further developed to improve procedural modeling and urban design, aiming to provide an effective means for quick architecture creation. These efforts have focused on the following four areas:

(1) **Layout modeling** refers to the procedural generation of parcels inside city blocks. Several pieces of research aimed to synthesize new urban layouts by creating and/or joining fragments of pre-existing examples [22–26]. For instance, Aliaga et al. [22] performed both a structure-based synthesis and an image-based synthesis to create urban layouts using example fragments from several real-world cities. Vanegas et al. [25] presented skeleton-based subdivisions and oriented bounding box subdivisions to generate spatial configurations of parcels with high similarity to those observed in

real-world cities. In contrast, our method generates urban layouts through a set of procedural rules rather than reproduces or starts with existing urban layouts.

(2) **Building modeling** addressed the problem of generating 3D building models. Most of the previous works focused on generating a compact, efficient, and reusable procedural representation to construct a new 3D architectural model that resembles the original [27–31]. Aliaga et al. [27] proposed a method to construct a grammar from photographs, enabling the rapid sketching of novel architectural structures in the original style. Demir et al. [14] converted an architectural model into a split tree and synthesized new geometric models with the extracted split tree. Compared to previous work, the 3D architectural model generated from our workflow is land-use-based. That is to say, instead of reproducing a specific architectural style, we try to restore the general building types of the different urban functional zone.

(3) **Facade modeling** applied a segmentation algorithm for facade reconstruction from image data or LIDAR scanning results [32–37]. Van Gool et al. [36] discussed different facade reconstruction algorithms and used one rule set for the reconstruction of different kinds of buildings. Wan and Sharf [37] presented a method for finding the best segmentation of facades through maximum likelihood formulation and then reconstructing building facades from LIDAR scans using a grammar-based segmentation algorithm. In this work, we seek to explore the procedural representation method of facade modeling and texture mapping. The main difference from previous work is that we try to implement a land-use-based texture mapping.

(4) **Urban modeling** explored an efficient way of urban scale modeling and rendering [38–42]. Kuang [41] implemented highly memory-efficient modeling and rendering of urban buildings by proposing a hierarchical grid-based modeling method and a data structure called Non-Uniform Texture. Marvie et al. [42] introduced GPU Shape Grammars for real-time generation, tuning, and rendering of procedural models. In this work, we applied the CGA grammars for large-scale procedural modeling with the advantage of avoiding explicit storage of expanded geometry and delaying the generation of buildings until the rendering stage.

## 3. Methods

This work proposed a series of CGA rules to provide a complete workflow of semi-automated 3D urban design, including layout modeling, building modeling, and facade modeling process. The project was conducted in ESRI CityEngine software which uses CGA as the main scripting language to define rules of 3D content creation. The methodological approach used in this project is summarized in Figure 1.

### 3.1. Data Preparation

A geographical information system (GIS) based dataset was collected for the construction of urban models, including satellite terrain imagery, road centerlines, and 2D zoning plan polygons (Figure 2). Satellite terrain imagery provides basic geographic information about the study area. All urban elements were placed on the terrain map to model actual land-specific topography. Road centerlines, along with street width attributes, were used for road creation among blocks.

Two-dimensional zoning plan polygons come with some key parameters, including zoning usage, floor area ratio (FAR), building coverage (BC), maximum permitted height (Hmax), and setback parameters. Zoning usage defines the land use type consisting of built-up areas (residential, commercial, industrial institution, educational institution, infrastructure, and utilities) and unbuilt areas (agriculture, forest, bare land). The FAR is the ratio of a building's total floor area to the size of land upon which it is built. BC defines the maximum area that building footprints can cover on the plot surface. Hmax is the maximum permitted height of the construction in meters. Setback refers to the minimum distance between buildings and the lot line. These key parameters were used as inputs for the generation of the 3D urban plan.

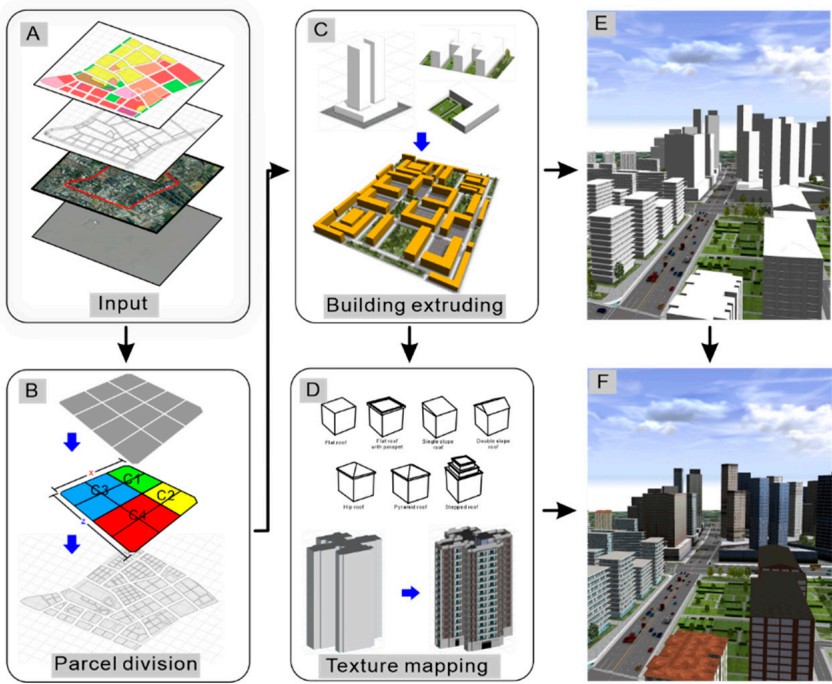

**Figure 1.** The workflow of semi-automated procedure modeling on urban scale, including parcel division, building extruding, and facade texture mapping process.

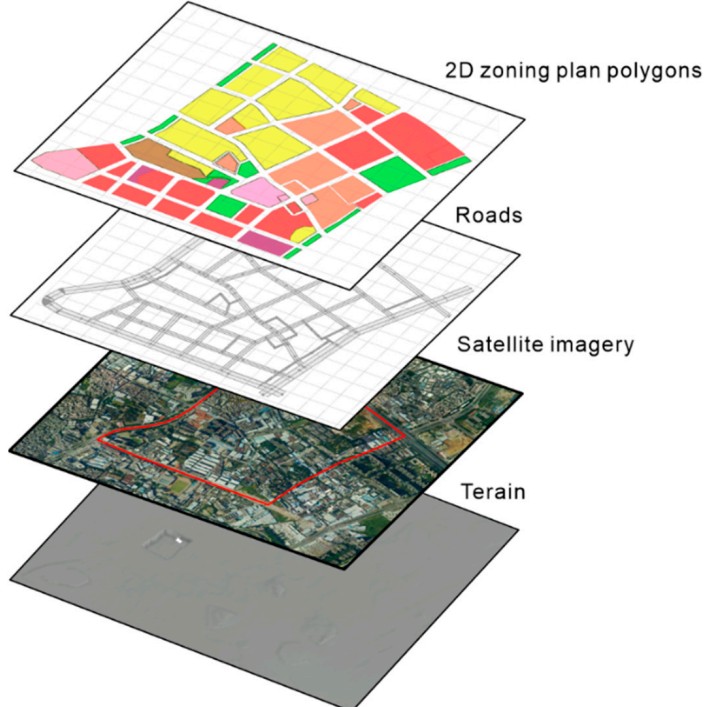

**Figure 2.** Dataset collected for the construction of urban models, including terrain with satellite imagery, road centerlines, and 2D zoning plan polygons.

### 3.2. Grammar Parsing and Derivation

CGA is a context-free grammar that can be written as

$$G = <T, \ N, \omega, R> \tag{1}$$

where $G$ is the grammar, $N$ is a set of non-terminals $N = \left\{ N_1 \cdots N_{|N|} \right\}$, representing the intermediate state of shape transformation, and $T$ is the set of terminals $T = \left\{ T_1 \cdots T_{|N|} \right\}$, representing outputs of 3D models. $\omega$ is a starting axiom, and $R$ is the collection of grammar rules, where the paradigm can be defined as

$$\langle N \rangle ::= \left[ \langle O_p \rangle \right] (\langle N \rangle | \langle T \rangle) \{ \langle N \rangle | \langle T \rangle \} \tag{2}$$

The output $N$ is the result of transformation and combination of a series of grammar rules from left to right. $O$ is the parameterized operator, including 11 basic transformation operations, i.e., extrude, split, rotate, comp, setback, insert, scale, transformation, L-shape, U-shape, and O-shape.

CGA grammar parsing obtains the general rules and terminals for 3D urban design. These rules describe the basic methods of: (1) parcel subdivision and clustering, (2) building extrusion, and (3) texture mapping. The process of deriving a completely urban design entails determining which production rules to apply and how many times to repeat them. The details are elaborated on below.

### 3.3. Urban Procedural Modeling

3.3.1. Parcel Subdivision and Clustering

Parcel subdivision and clustering determine the layout of buildings within each parcel. The subdivision of parcels to lots is carried out with one of the four algorithms: recursive, offset, skeleton, or partition (Figure 3). The first three methods are provided by the CityEngine. The recursive and offset subdivision algorithms create two types of lots: with street access or located in the middle of a parcel without street access. The skeleton subdivision algorithm can ensure that the created lots always have one side connected to a street (Figure 3A–C).

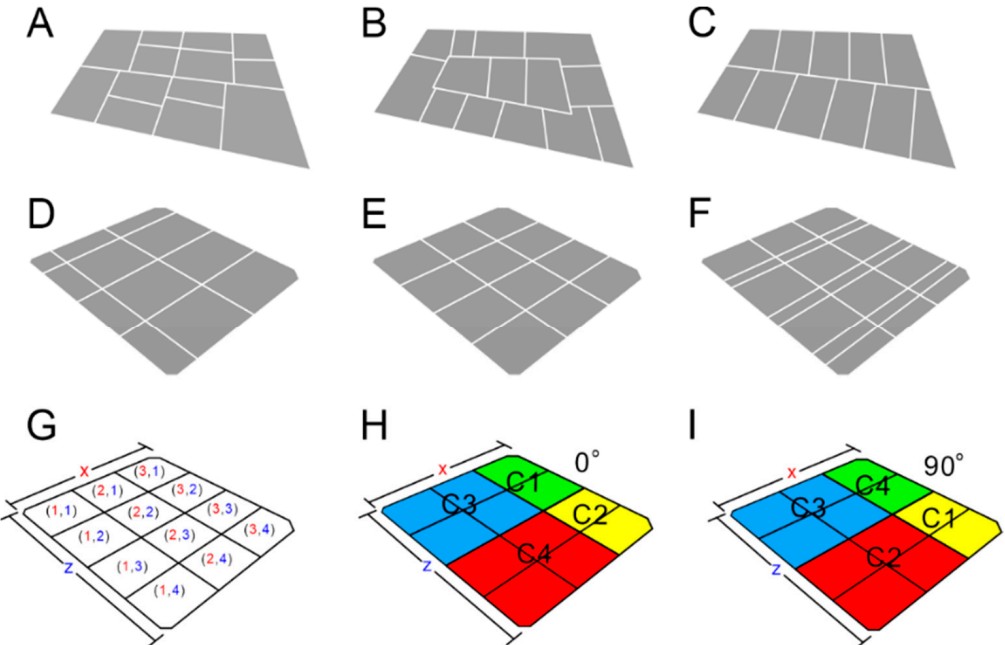

**Figure 3.** Parcel subdivision, clustering, and rotation ((**A–C**) showing subdivision results generated by recursive, offset, and skeleton algorithm; (**D–F**) showing subdivision results generated by partition algorithm with absolute values, specified partition number, and specified ratio; (**G**) showing index of inner lots, (**H**) showing inner lots clustered based on their index, (**I**) showing cluster lots rotating as a whole).

The partition algorithm proposed in this work enables subdivision of parcels into regular grids with a series of CGA rules. Starting from the 2D zoning plan polygons, a series of production rules *N* determines layout generation by combining the splitting and clustering operation. Firstly, 3 types of splitting methods are defined (Figure 3D–F). The type 1 rule specifies lot size as absolute values. The remaining parts at the edge of parcel remain the same (Figure 3D). The type 2 rule splits the geometry into a specified number of parts. This method creates equal size shapes with adaptive size (Figure 3E). The type 3 rule creates a repetitive pattern on the plot using the ratio (Figure 3F).

The clustering operation is then performed to create pre-defined layouts after splitting the parcel into smaller inner lots. Splitting rules conducted in the parcel subdivision step create an index system that can be used as the basis of the cluster operation. Starting from a certain corner, each inner lot is assigned an identifier index as "X index, Y index". Clustering rules pick out specific lots based on their index and assign them as a group. Normally, each group contains at most four lots. Defined clusters can be rotated and transformed as a whole to adapt the general intended layout of the plot (Figure 3G–I).

Parcel subdivision, clustering, and rotation are crucial in terms of layout design quality and diversity. Having the ability to choose different subdivision methods, define clusters, and rotate clusters gives the modeler flexibility to adjust the placement and size of the inner building lots.

### 3.3.2. Building Extrusion

With parcel subdivision completed, various building types are extruded from pre-defined layouts to generate different urban functional zone. All building extrusion rules are assumed to be organized in the following manner: (1) a pre-defined layout generated from above step is selected, (2) a production rule with this non-terminal as predecessor is chosen, (3) the non-terminal is replaced with the rule's successor to create the shape transformation result (e.g., a typical building of a selected zoning plan), and (4) the production process is terminated if all non-terminals are substituted; otherwise, the process repeats from step 2.

This organization defines a hierarchy of production rules where the grammar terminals consist of 8 common building types and public facilities (e.g., parking lot and park). These 8 building types are main hall with podium, multi-towers with podium, building with connecting corridor, tower building, O-shape building, L-shape building, U-shape building, and I-shape building (Figure 4).

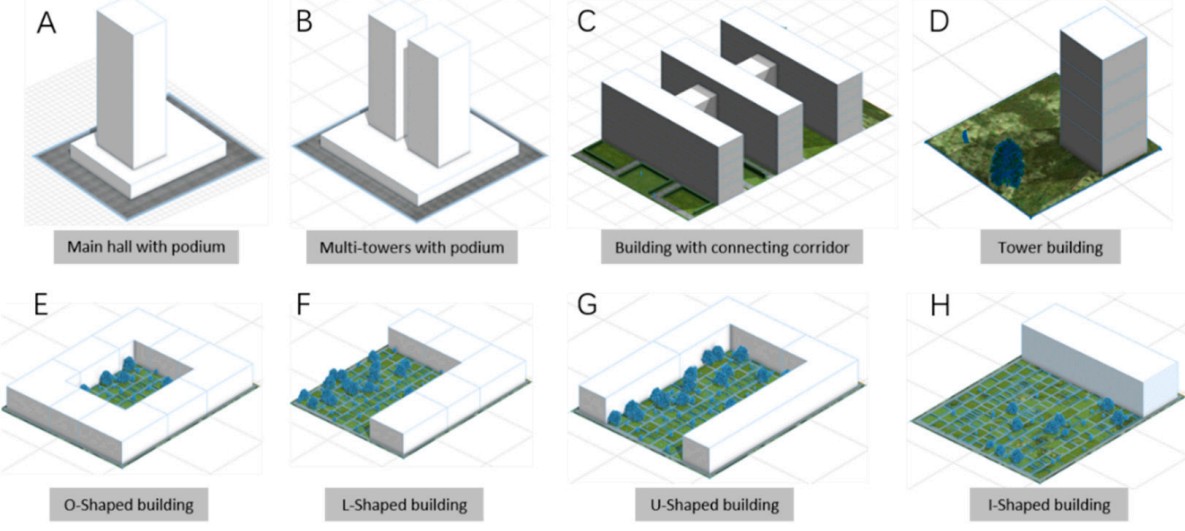

**Figure 4.** Building library containing 8 common building types ((**A**). Main hall with podium, (**B**). Multi-towers with podium, (**C**). Building with connecting corridor, (**D**). Tower building, (**E**). O-shape building, (**F**). L-shape building, (**G**). U-shape building, and (**H**). I-shape building).

A land-use-based combination of building types and layout is proposed subsequently to generate various urban forms of different urban functional zones (Figure 5). Urban form, referring to the size, shape, and configuration of an urban area or its parts, is determined, at a broad scale, by the building type, street type, and their layout [43]. The commercial land use zone contains three typical urban forms. The retail park often includes 4 L-shape buildings placed on a clustered parcel layout. A square, parking lot, or courtyard is placed in the center without street access. Shopping malls are mainly high-rise buildings (i.e., types I and II) in the street accessible parcels generated by recursive, offset, and partition algorithms. The pedestrian mall is characterized by dense buildings arranged in parallel partition parcels to form a narrow street space.

| Functions/ Land Use | Type | Description | Pattern | 3D Diagram |
|---|---|---|---|---|
| Commercial | Retail Park | Building Cluster with a square or courtyard in the center | | |
| | Shopping Mall | Building complex distributed in a compact way | | |
| | Pedestrian Mall | Buildings arranged in parallel to form the street space | | |
| Official | U-Shaped Building Clustering | Building Cluster with a square enclosed on three sides | | |
| | L-Shaped Building Clustering | Building Cluster with a square enclosed on two sides | | |
| | Office Buildings with atrium | Building complex distributed in a compact way | | |
| | Single Office Buildings | Buildings arranged in parallel | | |
| Residential | Slab Block | Buildings arranged in parallel | | |
| | Tower Apartment | Tower buildings arranged in parallel to increase outdoor space on the site | | |
| Industrial | Industrial Park | Buildings arranged along the loop | | |
| Educational | Kindergarten /Primary School | Buildings with connecting corridor sitting near a playground | | |

**Figure 5.** Various urban forms of different urban functional zone generated by land-use-based combination of building types and layout.

The official land use zone consists of four typical urban forms. The U-shaped cluster often includes two facing L-shaped buildings and two facing I-shaped buildings. It forms a square enclosed on three sides. Similarly, an L-shaped cluster brings out an L-shaped layout on a corner, and two I-shaped layouts on cross-sides, forming a square enclosed on two sides. Office buildings with atriums are characterized by a series of O-shaped buildings distributed in a compact way, and single office buildings are often arranged in parallel partition parcels.

The residential land use zone mainly consists of panel-type and towel-type building footprints (i.e., slab block and tower apartment). Both of them are arranged in parallel partition parcels. The industrial land use zone is normally dominated by simple basic building footprints (e.g., L-shaped, U-shaped, and I-shaped buildings). The educational land use zone, on the other hand, is characterized by buildings with connecting corridors.

The building heights in each zone are determined using the FAR, building density, and floor height. They can be calculated as $H = FAR * f_h/d$, where $H$ represents building height, $f_h$ is floor height and $d$ represents building density. By adding a random function, the building height variation is created, while the overall FAR remains unchanged. The setback parameter is used to fine-tune street configuration and design.

### 3.4. Texture Mapping

A typical building consists of several floors, each floor is divided into various faces, and each face consists of several windows surrounded by trim and wall material. Our algorithm exploits this typical structure to add details on extruded building footprint and then map texture from image space to 3D architecture model space. In this process, texture maps and grammar rules are compiled together as buffer objects to support efficient access by fragment shaders. Each pixel that needs to be colored follows the decision tree constructed by the grammar to derive the texture coordinates.

For parsing, all buildings are divided into ground floor, intermediate floors, and roof from bottom to top (Figure 6). There are seven types of roofs: flat roof, flat roof with parapet, single slope roof, double slope roof, hip roof, pyramid roof, and stepped roof. Slope, hip and pyramid roofs are often observed on residential buildings, while other roof types can be observed on all kinds of buildings (Figure 7).

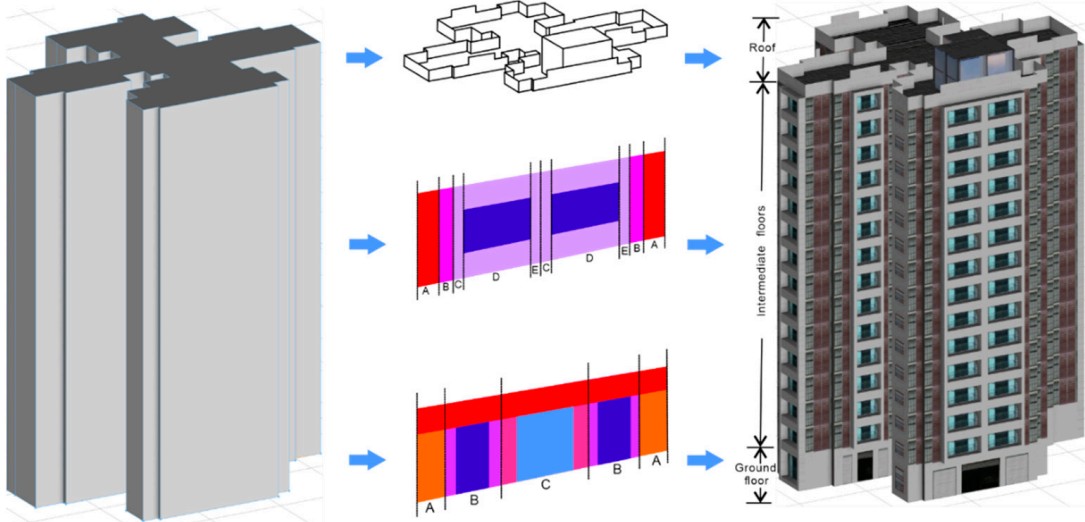

**Figure 6.** Texture mapping of building, dividing a typical building to ground floor, intermediate floors, and roof from bottom to top; Each part consists of repeating pairs of elements; The detailed facade synthesis is conducted by combing the grammar-based splitting with texture mapping.

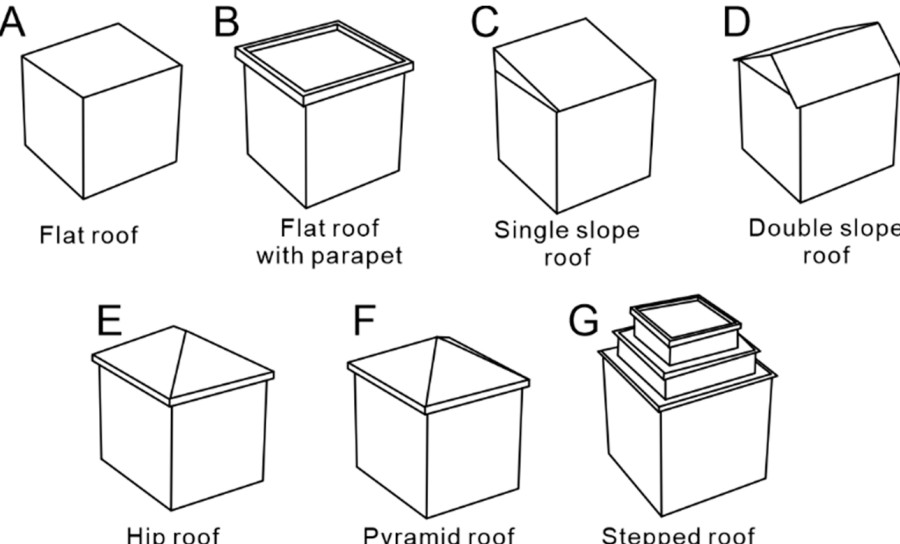

**Figure 7.** Seven types of roofs ((**A**) Flat roof, (**B**) Flat roof with parapet, (**C**) Single slope roof, (**D**) Double slope roof, (**E**) Hip roof, (**F**) Pyramid roof, and (**G**) Stepped roof).

Although ground floors and intermediate floors normally have different structures and textures, they can be decorated in two ways. The first method is the creation of texture maps representing each planar section of a building's facade. The second is decomposition of the facade by a set of repeating elements. The ground floor and intermediate floors consist of an ordered sequence of facades. Each facade can be subdivided into labeled feature groups representing brick, trim, windows, and entries. The subdivision process can be summarized as a facade-splitting schema, including the symbolic growth rule and geometric properties. Reliably discovering the underlying symmetry in structure and texture is crucial for facade analysis. For example, considering the face in Figure 6, which can be subdivided into 10 columns, if A represents brick and B corresponds to a window, then the pictured face has only 2 unique types of columns and can be written as $F = ABCDECDEBA$. Considering that the repeating pairs of elements can be combined and represented by borrowing the Kleene star notation from regular expression, the facade splitting rule is rewritten as $F = AB(CDE)^*BA$ (Figure 6).

CityEngine provides a standard texture library that can be used to map appropriate texture images to the matching building facade based on land used type and building height. However, to generate a city block with specific local characteristics, a customized texture library is required. Since the texture mapping process of building facade involves the transformation from image space to building model space, strict constraints on texture quality are proposed: (1) all texture images should be taken with orthorectified view since oblique view often results in a skewed facade texture; (2) all texture images should be regular shape since irregularly shaped polygons often cause geometric distortion; (3) all texture images should be relatively clean since occluded elements often cause erroneous texture mapping of the building model.

In this work, a rudimentary method proposed by Früh and Zakhor [44] has been employed to perform local texture image acquisition. A series of post-processing techniques, e.g., foreground removal, geometric correction, and color balancing, proposed by previous researchers [45–47], was also applied to improve the quality of texture images. Thus, a land-used-based texture library containing six types of facades (commercial, official, residential, industrial, educational, infrastructure and utilities) and four types of basic facade elements, including windows, walls, doors, and roofs, is constructed (Figure 8).

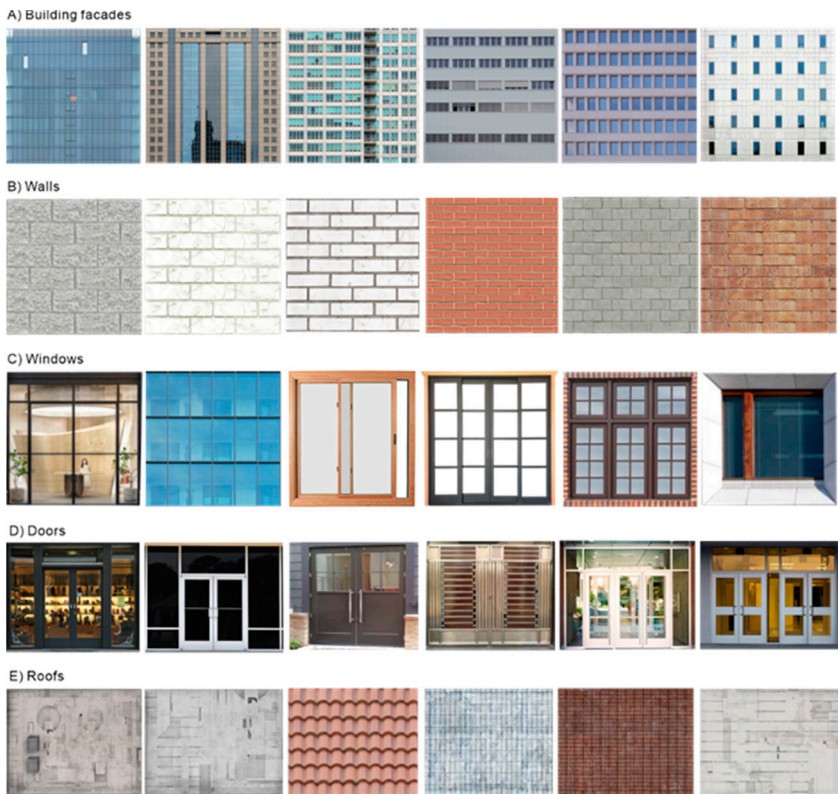

**Figure 8.** A land-used-based texture library. ((**A**) Six types of building facades, namely commercial, official, residential, industrial, educational, infrastructure, and utilities; (**B–E**) showing 4 types of basic facade elements, namely windows, walls, doors, and roofs).

## 4. Case Study

To demonstrate the proposed urban procedural modeling described and explore future potential, it was used to generate and optimize the urban design in two practical projects in Guangzhou, a megacity in south China. Both projects are covered by a detailed control plan, and each has its own characteristic. One is the urban renewal project of H-Village, involving various stakeholders, namely the village manager, peasants, local community, residents, developers, local government, and so on. In this project, our methodology shows the capacity to coordinate and collaborate with multiple stakeholders by providing a flexible urban design proposal that can be modified on the fly. Another is an urban design strategy for the air–rail integration zone, which is planned as the major transportation hub of Guangzhou. In this project, our methodology generates multiple urban design proposals to cover possible urban development scenarios in order to fully consider the uncertainty existing in the long-term development of the city.

*4.1. Case 1: Urban Renewal Project of H-Village*

4.1.1. Background

H-Village is located in an urban–rural fringe of Guangzhou, China, 11 km north of the city center (Figure 9A). The project covers an area of 120 hectares. It can be seen from the satellite map that there are rural communities, town communities, and industrial zones on the site (Figure 9B). In 2018, the local government raised a conceptual plan called Guangzhou Valley of Design to create a hub for design enterprises. After years of negotiation, 90% of the existing residents agreed to the urban redevelopment proposal in early 2021. The zoning adjustment begins soon according to the new development strategy (Figure 10), as well as the demolition.

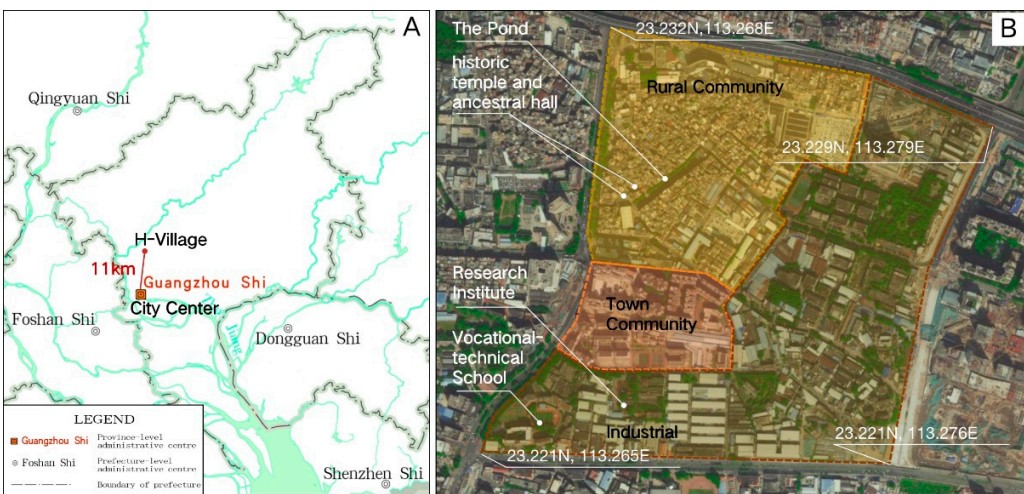

**Figure 9.** Location and current situation of H-Village ((**A**) showing the location of H-Village, 11 km north of the city center, and (**B**) showing the current situation of H-Village, consisting of rural, town, and industrial communities, with some reserved buildings and landscape in the area).

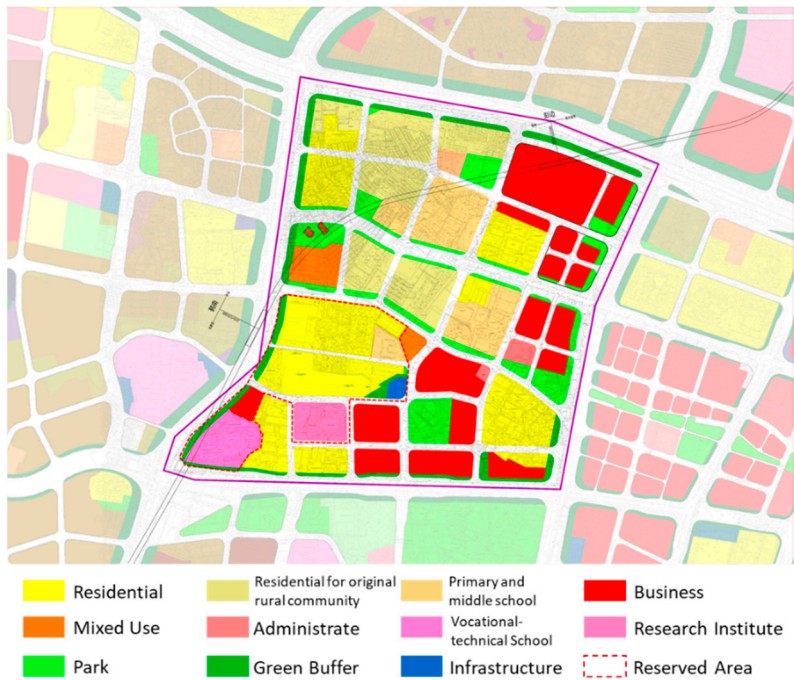

**Figure 10.** The detailed plan of the H-village redevelopment project after zoning adjustment (the newly built part consisting of 13 parcels of residential, 4 parcels of educational, and 15 parcels of business).

In the latest detailed plan, parts of the existing functions are reserved, namely the town community, a historical temple, an ancestorial hall, a vocational-technical school, and a research institute, while the rural community will become a modern community consisting of apartments, schools, parks and public facilities, and the industrial zone will be developed into a creative zone to offer jobs and homes. The newly-built part, namely the former rural community and the industrial zone, mainly consists of thirteen parcels of residential, four parcels of educational, and fifteen parcels of business (Figure 10).

### 4.1.2. Restrictions and Requirements

In this case, restrictions and requirements are from different groups of stakeholders, namely residents/peasants from rural communities, residents from town communities, the owner and manager of the industrial zones, the local government authority, the developer, Guangzhou Metro, and enterprises which intend to settle in Table 1. There are some interests all the stakeholders share, such as accessibility to amenities, appropriate open space, and energy saving. There are some special requirements as follows.

**Table 1.** Restrictions and requirements of different groups of stakeholders in the H-Village project.

| Group | Restrictions and Requirements for the Urban Design |
| --- | --- |
| Rural Community | Proper ratio of the residential function<br>Keeping the cultural context |
| Town Community | Keeping the existing community<br>Control of height |
| Guangzhou Metro | Sparing open space for Line 14 and stations |
| Enterprises | Identity of the site/Fantasy facade<br>Network of open space |
| Developer | Financial balance<br>Feasibility |
| Local Government Authority | Street view<br>Future development<br>Satisfaction of residents |

Most of the residents from the rural community will come back to settle down after the redevelopment. They would argue for enough living places and care about the cultural context, which leads to the proper ratio of the residential function and a proper design solution for reserving the little pond, historic temple, and ancestral hall.

Since the town community will be reserved as before, residents from the town community require sufficient daylight and sunlight for the existing community when new buildings rise from the ground in the south. Thus, designers should take the distance to the existing community and the height of the new building into consideration.

According to the metro plan, Line 14 will pass through the site in the northwest corner, and two stations of Line 14 will be located at the site. Guangzhou Metro, which is responsible for the construction and operation of Line 16, requires that there should be open space on the ground above Line 14 to provide enough space for near-future construction. The stations could be attached to surrounding buildings or stand alone.

Enterprises intending to settle in are mainly in the field of design, including industrial design, architecture design, graphic design, and fashion design. As future users, they may prefer a fantasy facade to demonstrate innovative culture and continuing open space to make chance encounters take place.

Both the developer and the local government authority are looking forward to the comprehensive performance of the urban design proposal, as they are officially in charge of the entire project and the future operation. However, they focus on different aspects. The developer will concentrate on the financial balance and feasibility problem, while the local government authority concerned with the street view, future development, and satisfaction of the residents.

### 4.1.3. Multiple Proposals Generation

After preparing the data and conducting the first two steps in the CGA rules, some alternative proposals were raised for the project. Three of them were chosen to focus on how they correspond to the restrictions and requirements. The main design issues in the H-Village project are to reserve certain parts in the site, to make the newly built parts work with the reserved parts well, and to construct a new identity in the redevelopment project.

The site could be divided into reserved zones, zones adjacent to the reserved zones, and newly built zones (Figure 11). The reserved zones are where the reserved constructions or landscapes exist, while the other two zones are mainly newly built. The zones adjacent to the reserved zones are influenced strongly by the reserved construction and the existing community, mainly consisting of residential parcels and educational parcels. The newly built zones are the main component of the skyline and the road landscape.

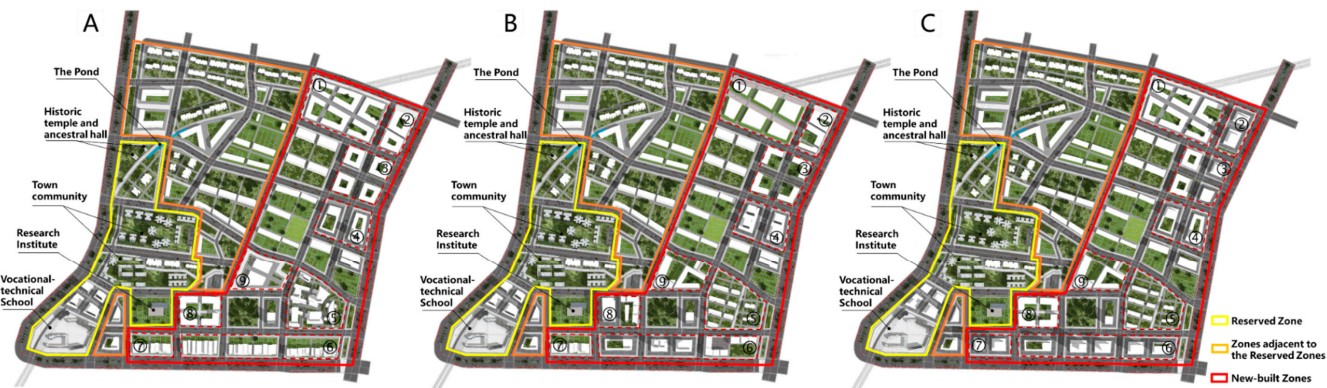

**Figure 11.** Three alternative proposals in the H-Village project ((**A–C**) sharing the same reserved zones and the zones adjacent to the reserved zones but differing in the newly built zones, mainly consisting of business parcels and residential parcels, especially Blocks 1–9).

All three proposals share similar reserved zones and the zones adjacent to the reserved zones. The reserved parts are mainly in the west and southwest, including the pond, a historic temple, an ancestral hall, the town community, the research institute, and the vocational-technical school. They are identified and extruded to reflect their actual appearance using the terrain with satellite imagery. The same pattern languages are applied in zones adjacent to the reserved zones, as it is the optimal solution when considering the orientation, FAR, proper open space, setback, and other limitations set by zoning.

To protect and continue the contexts shaped by the culture, some green spaces and routes are spared for the residents/peasants from rural communities and the newcomers. The residential parcel at the east of the historic temple employs the tower apartments to avoid obstructing the view, while the other residential parcels utilize slab blocks to meet the living requirements of residents/peasants (Figure 11).

The differences in the three proposals are the patterns in newly built zones, consisting of business and residential parcels, which form the identity of the site and are mainly decided by the design concept of street interfaces in every proposal. Specifically, these parcels are divided into nine blocks, labeled from 1 to 9 clockwise (Block 1 along the northern boundary, Block 2 at the northeast corner, Blocks 3–5 along the eastern boundary, Block 6 at the southeast corner, Block 7 along the southern boundary, Block 8 north of Block 6, and Block 9 northeast of Block 8).

Plan A is the lowest density proposal in response to the residents' demands for living places and sunlight. It creates a lot of open space through the enclosure and semi-enclosure forms. In this case, the blocks along the street are of intermediate height and have certain patterns with complete street fronts. To be more specific, they are Block 1 with U-shaped business buildings, Blocks 2–3 with O-shaped business buildings, Block 4 with the main hall with a podium, Block 5 with residential building complex, Block 6 with single office buildings, Block 7 with slab blocks, Block 8 with building clusters with a courtyard in the center, and Block 9 with a compact layout.

Proposal C could be seen as a more identical solution, revealing the government's ambition. Almost every parcel along the main streets features tower buildings or tower buildings with podiums, namely Blocks 2–4 and Blocks 6–7. Blocks 1 and 5 in proposal

C take their patterns from proposals A and B, respectively. There are many spare spaces around the tower buildings, making it possible to implant a network of various open spaces.

Compared to proposals A and C, Plan B is a compromised option. There is a desire to create an identity contrary to the reserved parts behind the road interface in proposal B, so some tower buildings emerge in Block 6 to the south and Block 4 to the east. These tower buildings are potential landmarks. The other parts along the street are moderate with some varieties; for example, Block 1 applies a compact layout, and Block 2 combines U-shaped and L-shaped business buildings.

#### 4.1.4. Alternative Facade Proposals

Since enterprises are looking forward to a fantasy streetscape, there are several alternative facade proposals in the multiple proposals mentioned above (Figure 12). Colors, transparency, and styles could indicate the functions of buildings, such as offices (Figure 12A,B), schools (Figure 12C), and businesses (Figure 12D). There are three common textures for modern office buildings (one in Figure 12A, two in Figure 12B). The first consists of white walls and blue glass, in which the glass is fragmented following certain modules to express an erect and exquisite image. The second is decorated by pale brown ceramic tiles and ribbon windows. All the office rooms can receive equal light with the ribbon windows, so the facade is applicable for research or design institutes. The third one applies glass curtain walls to make the office building appear to be light and clear, which will be popular with creative classes. In the typical facade for schools, the window height is much larger than others to make natural light penetrate deeper into the building. Every classroom could have a good light condition with three or four windows. Figure 12D shows a street view with a business facade texture along the street. On the left is the ground floor of a business building, where retail activity could take place, such as cafés and grocery stores, while on the right is a vital texture with irregular segmentation. The same functions can also use various different facade textures. There are several visual effects when applying certain textures in the same building (Figure 12E–H). Figure 12E,F employs double slope roofs, while Figure 12G,H utilizes flat roofs. Aside from the roofs, they vary in wall materials, including brown bricks, brownish-red ceramic plate curtain walls, brown ceramic tiles, and khaki bricks.

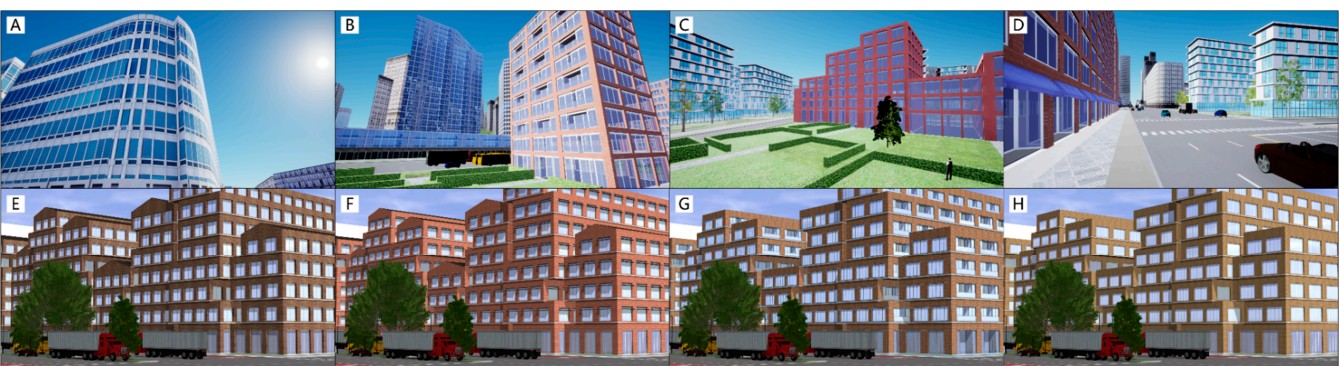

**Figure 12.** Different facade textures applied to the buildings ((**A**,**B**) showing textures for offices, (**C**) showing textures for schools, (**D**) showing textures for retail space, (**E**–**H**) showing different visual effects when applied certain textures in the same building, (**E**) applying double slope roofs and brown bricks on walls, (**F**) applying double slope roofs and brownish red ceramic plate curtain wall, (**G**) applying flat roofs and brown ceramic tiles on walls, (**H**) applying flat roofs and khaki bricks on walls).

#### 4.2. Case 2: Urban Design Strategy of Air–Rail Integration Zone

#### 4.2.1. Background

The air–rail integration zone is adjacent to the railway station of the Beijing–Guangzhou Railway, 28 km north of the city center and 11 km west of the airport (Figure 13A). The

project covers an area of 1081 hectares, almost 10 times the area of H-Village. There are rivers running through the site, making it an outstanding place with a natural environment. However, there are currently few rural communities and industrial zones along the rivers (Figure 13B). With the integration plan of the airport and the railway station proposed in 2020, an air–rail line is to pass through the site, leading to opportunities for new development in the nearby area.

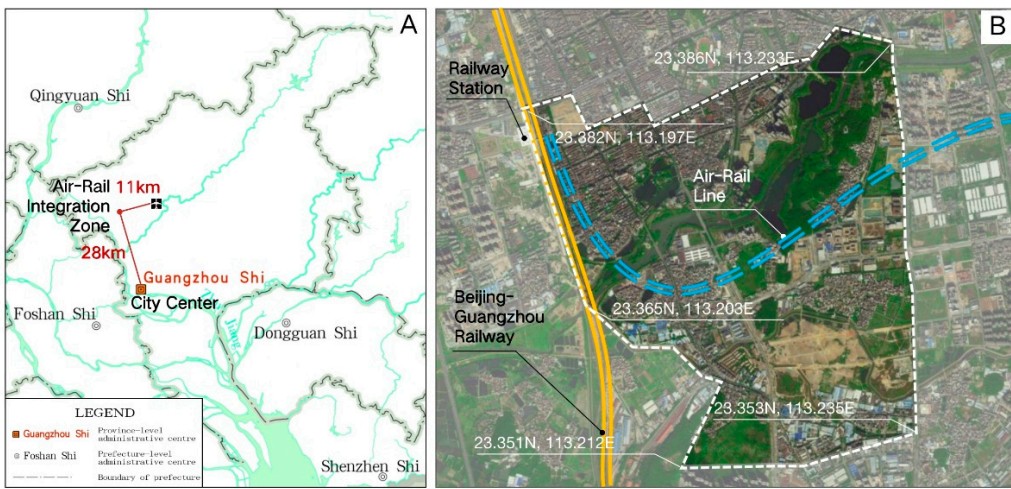

**Figure 13.** Location and current situation of air–rail integration zone ((**A**) showing the location of site, 28 km north of the city center and 11 km west of the airport, and (**B**) showing the current situation of the site beside the Beijing–Guangzhou Railway, consisting of rural and industrial communities, with the future air–rail line passing through).

There are five clusters in the existing detailed plan. They are divided by rivers and main roads (Figure 14). Clusters 1–3 are the core clusters along the air–rail line. Cluster 1 is next to the Railway Station with small blocks of business and commercial and residential areas. Cluster 2 is on an island with surrounding water. It applies a radiate road system; thus, there is a visual focus area around Y-Station. Cluster 2 is the most comprehensive cluster among the five, including business, business for innovation, administration, and commercial and residential areas. Cluster 3 is an innovation zone with large parcels of business for innovation. Cluster 4 is north of Clusters 1–3, consisting of residential areas and other amenities, while Cluster 5 is south of Clusters 2–3, consisting of business, business for innovation, and residential.

### 4.2.2. Restrictions and Requirements

Rivers are the main natural elements in the case. To prevent flood disasters, it is necessary to apply green buffers of a certain distance around every cluster. The rivers make the site unique and provide an outstanding environment, making the area more suitable for a landscape system.

The entire integration zone is located in the special customs supervision zone; therefore, the design must meet the state's technical standardized requirements of airport clearance protection, namely 165 m. In that case, the height of the construction plus the ground altitude is less than or equal to 165 m. Since the ground altitude varies in every parcel, it is necessary to set the height control carefully (Table 2).

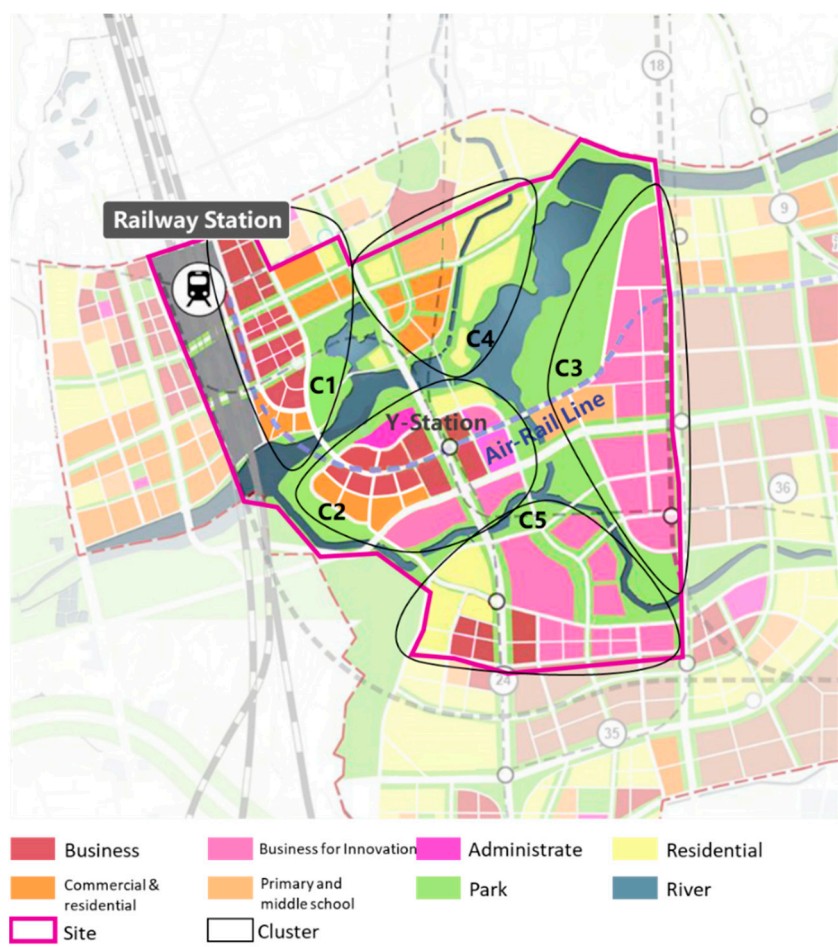

**Figure 14.** The detailed plan of the air–rail integration zone project (five clusters near rivers and main roads; cluster 1–3 along the air–rail line, mainly consisting of business, business for innovation, and commercial and residential areas; cluster 4 north of cluster 1–3, consisting of residential areas and other amenities; and cluster 5 south of cluster 2–3, consisting of business, business for innovation and residential areas).

**Table 2.** Restrictions and requirements in air–rail line project.

| Elements | Restrictions and Requirements for the Urban Design |
|---|---|
| Rivers | • Green buffer along the rivers<br>• Proper road system and landscape system |
| Special Customs Supervision Zone | • Control of height |
| Air–Rail Line Development | • Control of density |

In terms of air–rail line development, there should be appropriate density development in each cluster. The station traffic is one of the most important factors influencing the density, especially in Clusters 1–3. When it comes to Clusters 4 and 5, the density should comply with the facility capacity (Table 2).

4.2.3. Multiple Proposals Generation

When focusing on the main design issues, the parcels in Clusters 1–3 along the air–rail line are the main differences in the three alternative proposals since the green buffer and

height control are the same, and the facility capability is stable in Clusters 4–5. The forms of construction in Clusters 1–3 reveal different concepts to deal with the density issue and landscape system (Figure 14).

We present three alternative proposals for the air–rail line project in Figure 15. Proposal A (Figure 15A) is set for a high-traffic need situation in both the railway station and Y-Station, so it utilizes a high-density development and centralized landscape system. Specifically, tower buildings are the main form, no matter the business parcels in Cluster 1, residential parcels in Cluster 2, and some business for innovation parcels in Cluster 3, maximizing the density to the limit of the FAR. Since Cluster 3 is beyond walking distance from Y-Station, only the core district of Cluster 3 takes up the tower building form, while the other parts consist of U-shaped and O-shaped buildings. The building complex that stretches to the river and the surrounding parcels in Cluster 2 forms the focal point of the site, and the nonlinear architecture near the river in Cluster 1 acts as a landscape node, together making the centralized landscape system.

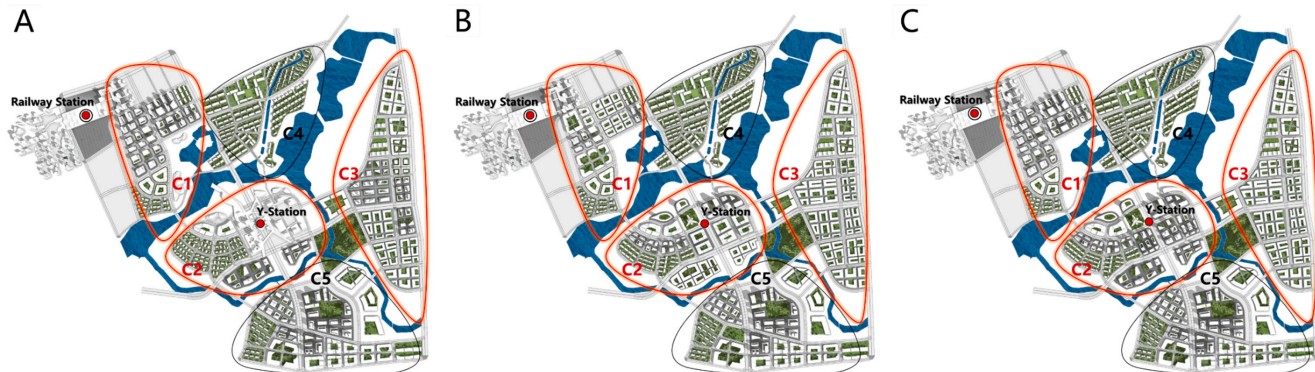

**Figure 15.** Three alternative proposals in the air–rail line project (Proposal (**A**) employing a high-density development and centralized landscape system in a high traffic need situation in both the railway station and Y-Station, Proposal (**B**) employing a low-density development and decentralized landscape system in a low traffic need situation in both the railway station and Y-Station, Proposal (**C**) employing a medium-density development and decentralized landscape system when there is a high traffic need at the railway station and a medium traffic need in Y-Station).

On the contrary, Proposal B (Figure 15B) aims to address the low traffic need situation in both the railway station and Y-Station. More O-shaped and U-shaped buildings are employed to adapt to a low-density development and decentralized landscape system. The residential parcels in Clusters 1 and 2 take the form of low-rise slab blocks or O-shaped buildings, and the business for innovation parcels in Cluster 3 utilize the low-rise U-shaped buildings. When it comes to the landscape system, there are courtyards of various sizes in Cluster 1; thus, every single building could face a semi-private green space beside the river scene. Instead of the attractive building complex in Cluster 2 being the center of the landscape system, there are landscape sequences from the riverside and green spaces beside the station to the courtyard in O-shaped or U-shaped buildings, making the decentralized landscape system.

Proposal C (Figure 15C) could be seen as an intermediate state between Proposal A and B when there is a high traffic need at the railway station and a medium traffic need at Y-Station simultaneously. In this case, Cluster 1 is the same as that in Proposal A, and Cluster 3 is the same as that in Proposal B. Since the passenger flow from Y-Station is expected to be lower than that in Proposal A and larger than that in Proposal B, in Cluster 2 there are fewer tower buildings than in Proposal A, but more than in Proposal B. As well as Proposal B, Proposal C employs a decentralized landscape system, but the sequences are simplified as two levels of the riverside and green spaces beside the station.

## 5. Discussion

Geometric modeling (GM), building information modeling (BIM), and procedural modeling (PM) are the three main approaches to urban design modeling. Among them, GM is the most widely accepted method by urban designers since it can realize the complete expression of the designer's ideas with a high degree of realism in urban modeling. Application of GM, however, is labor-intensive and time-consuming. BIM is often used in building design since it contains all the relevant information needed for the planning, construction, and operation of a building, but it is not a suitable tool for urban design since some key elements (e.g., roads and public facilities) are not included within BIM.

Compared to the aforementioned two methods, PM has two apparent advantages:

(1)    PM is the most efficient modeling method of urban design. With the well-developed CGA algorithm, designers can generate several square kilometers of urban design models in a matter of seconds, which allows urban designers to generate design models nearly real-time. Although the models produced by the PM method merely achieve similar degrees of accuracy to those produced by the BIM or GM method, the urban designer can easily use this method to complete the preliminary evaluation of different schemes in the early stage of urban design.

(2)    PM is capable of creating flexible geometries which can accommodate various boundary conditions. Relying on this ability, an agile urban design working process is applied, with the goal of allowing designers to take advantage of a high level of automation while still maintaining a high degree of control over the output. There are two working models: "generation" and "modification in the PM-based urban design process. A designer can easily switch between the two working modes. The generation model provides a fully automatic workflow that allows urban designers to generate a large variety of urban building models with the constraints of a detailed plan. The modification model gives urban designers options to modify selected model data according to different urban design concepts and needs to handle information that is not encoded into the automated processes.

In this PM-based method, an urban design process is divided into a series of steps, allowing designers to intervene and modify the outputs of each step which enhances collaboration between machines and humans. It ensures real-time communication between conceptualization, design, and operation. Stakeholders' engagement enables the improvement of the quality of services and of wellness in the environment. The approach proposed by this work can improves the automation level of urban design work significantly.

There, however, are two challenges that may limit its practical application: (1) The general algorithm does not cover all special circumstances that need to be dealt with. In order to effectively handle a variety of specific needs, researchers may need to add more custom work beyond the general modeling process, which significantly increases the complexity of algorithm development. In this circumstance, urban planners usually prefer traditional methods since algorithm development can be more complicated and time-consuming. (2) Quantitative assessment of urban planning indicators (e.g., land usability, planned building volume, traffic accessibility, green space accessibility, traffic noise impact), which would significantly reduce urban planners' assessments of the feasibility of planning options, is not covered in this approach. This will be implemented in future research to better assist planners in realizing interactive urban design work.

## 6. Conclusions

The aim of this study was to explore a 3D urban procedural modeling solution for urban designers to enhance collaboration among stakeholders with a computer-aided design tool.

A series of CGA rules addressing the needs of 3D urban modeling is proposed. Compared with the previous work, it involves a complete workflow of semi-automated design procedures at an urban scale, including layout modeling, building modeling, and facade modeling processes. This workflow was employed in the H-village urban redevelopment

program and the air–rail integration zone development program in Guangzhou. Three preliminary design proposals were generated for each project. The result demonstrated that this workflow could settle most of the collaborative issues with its analysis functions, flexible adjustment mechanism, and real-time visualization. Meanwhile, creative design issues can be dealt with well by generating multiple layout proposals and alternative facade textures quickly.

**Author Contributions:** Conceptualization, Ming Zhang and Jielin Wu; methodology, Ming Zhang; software, Ming Zhang; validation, Yang Liu, Ji Zhang, and Guanyao Li; formal analysis, Jielin Wu; investigation, Ji Zhang; resources, Ji Zhang; data curation, Ming Zhang; writing—original draft preparation, Ming Zhang and Jielin Wu; writing—review and editing, Yang Liu, Ji Zhang, and Guanyao Li; visualization, Jielin Wu; supervision, Yang Liu; project administration, Yang Liu; funding acquisition, Yang Liu. All authors have read and agreed to the published version of the manuscript.

**Funding:** This research was funded by Guangdong Enterprise Key Laboratory for Urban Sensing, Monitoring and Early Warning (No.2020B121202019).

**Data Availability Statement:** Not applicable.

**Conflicts of Interest:** The authors declare no conflict of interest.

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
