# Peer review of "GIS Based Procedural Modeling in 3D Urban Design"

_ijgi, doi:10.3390/ijgi11100531_

Round 1

Reviewer 1 Report

The authors propose a computer-generated architecture based workflow to be used in urban design, where it can support not only visualization but also collaboration and communication between involved parties. The workflow is based on 3D procedural modeling and consists of parcel subdivision and clustering, generating various layouts for different urban functional zones, building extruding and texturing their facades. The discussed case studies show that the proposed workflow can quickly generate multiple layout proposals.

The presented framework is promising and provides an interesting contribution to the field, however it can benefit from some improvements. There is a lot of research work done in this field and the originality of this paper should be stressed by describing the contribution in comparison to some of them (see point 2 in the comments).

 Some more detailed comments:

1)     Use of English:

- There are some language mistakes like ‘it still has tremendous advantages to have..’, ‘this work enables subdivided’, ‘Splitting rules … creates..’, ‘assign them as a group’, ‘different urban functional zone’, ‘Commercial land us’, ‘a series of O-Shaped building’, ‘Roof contains seven types’, ‘has employed to perform’, ‘if A’, ‘the tower buildings are less than…but more than’.

- You use both ‘facade’ and ‘façade’,

-        Moreover articles are often omitted (before ‘rule, front, shape, operation, layout, footprint, floor, roof, land, façade, polygon, space)

2) In Section 2 the references to such papers as:

a)          Jean-Eudes Marvie, Cyprien Buron, Pascal Gautron, Patrice Hirtzlin, Gaël Sourimant, GPU Shape Grammars, 2013

b)     Kuang et al., A compact random access representation for urban modeling and rendering, 2013

c)     Wang, Sharif, ‘Grammar-based 3D façade segmentation and reconstruction’, 2012

should be added and the presented approach should be compared to these works

3)     The caption under Fig. 2 is the same as the one under Fig.1.

Reviewer 2 Report

The manuscript entitled 'GIS Based Procedural Modeling In 3D Urban Design' describes a method for constructing 3D models at an urban scale. The aims of the article are clear and the images proposed by the authors are of high quality. However, the approach suggested by the authors is well suited for small areas; an application on an urban scale is difficult as it requires rather demanding semi-automatic procedures.

Below are some comments and suggestions, line by line.

Line 9       I suggest to change or delete the term “innovative”

Line 72     I suggest more recent reference describing the construction of 3D                   models, such as:

https://doi.org/10.1080/19475683.2022.2037019

https://doi.org/10.3390/ijgi10100697

http://dx.doi.org/10.28991/CEJ-2022-08-01-08

Line 109   I suggest to introduce the Methos section and then the fig. 1 - 2

Line 326   It is necessary to add the Country where it is located this area

Line 335  It is necessary to add the geographic coordinates if the site or                        introduce a geographic frame

Line 385   After fig. 12, the numeration of the page start from 1 …                                   please check and update the manuscript

Pag. 18   The Discussion section should be expanded by emphasising                          advantages and limitations of the method proposed by the authors                and comparing it with other methods.

Lastly, it is necessary to report in ascending numerical order the citations. Please check the references according the guideline of the journal. 

Reviewer 3 Report

This paper proposes a computer-generated architecture (CGA)-based workflow, which enhances collaboration amongst urban desiners and boosts their creativity with a computer-aided design tool.

Ths workflow consists of three steps, 1) parcel subdivision and clustering, 2) building extruding, and 3) texture mapping for buildings' facade. The authors introduce two projects carried out based on the proposed workflow, those are an H-village urban redevelopment project and an Air-rail intergration zone development project.

The contents of this paper may be interesting for urban desiners. However, the proposed workflow is very natural ,and there are not any theoretical nor technological innovative aspects. The contents of this paper looks the explanation of the highly sophisticated urban design software produced by ERIS. Therefore, this paper cannot be accepted as IJGI paper.

Round 2

Reviewer 1 Report

The authors have adequately addressed my previous concerns.

Reviewer 2 Report

The authors responded to all my comments and suggestions. Therefore, I have no further comments since the paper is significantly improved from the previous version.

Reviewer 3 Report

The revised paper satisfies my review comments. So, I do not have more any comments.